# TRAINING NEURAL NETWORKS FOR ASPECT EXTRACTION USING DESCRIPTIVE KEYWORDS ONLY

**Giannis Karamanolakis & Daniel Hsu & Luis Gravano**
Columbia University
New York, NY 10027, USA
`{gkaraman, djhsu, gravano}@cs.columbia.edu`

## ABSTRACT

Aspect extraction in online product reviews is a key task in sentiment analysis and opinion mining. Training *supervised* neural networks for aspect extraction is not possible when ground truth aspect labels are not available, while the *unsupervised* neural topic models fail to capture the particular aspects of interest. In this work, we propose a *weakly supervised* approach for training neural networks for aspect extraction in cases where only a small set of seed words, i.e., keywords that describe an aspect, are available. Our main contributions are as follows. First, we show that current weakly supervised networks fail to leverage the predictive power of the available seed words by comparing them to a simple bag-of-words classifier. Second, we propose a distillation approach for aspect extraction where the seed words are considered by the bag-of-words classifier (teacher) and distilled to the parameters of a neural network (student). Third, we show that regularization encourages the student to consider non-seed words for classification and, as a result, the student outperforms the teacher, which only considers the seed words. Finally, we empirically show that our proposed distillation approach outperforms (by up to 34.4% in F1 score) previous weakly supervised approaches for aspect extraction in six domains of Amazon product reviews.

## 1 INTRODUCTION

Aspect extraction is a key task in sentiment analysis, opinion mining, and summarization (Liu, 2012; Hu & Liu, 2004; Pontiki et al., 2016; Angelidis & Lapata, 2018). Here, we focus on aspect extraction in online product reviews, where the goal is to identify which features (e.g., price, quality, look) of a product of interest are discussed in individual segments (e.g., sentences) of the product's reviews.

Recently, rule-based or traditional supervised learning approaches for aspect extraction have been outperformed by deep neural networks (Poria et al., 2016; Zhang et al., 2018), while unsupervised probabilistic topic models such as Latent Dirichlet Allocation (LDA) (Blei et al., 2003) have been shown to produce less coherent topics than neural topic models (Iyyer et al., 2016; He et al., 2017; Srivastava & Sutton, 2017): when a large amount of training data is available, deep neural networks learn better representations of text than previous approaches.

In this work, we consider the problem of classifying individual segments of online product reviews to predefined aspect classes when ground truth aspect labels are not available. Indeed, both sellers and customers are interested in particular aspects (e.g., price) of a product while online product reviews do not usually come with aspect labels. Also, big retail stores like Amazon sell millions of different products and thus it is infeasible to obtain manual aspect annotations for each product domain. Unfortunately, fully supervised neural approaches cannot be applied under this setting, where no labels are available during training. Moreover, the unsupervised neural topic models do not explicitly model the aspects of interest, so substantial human effort is required for mapping the learned topics to the aspects of interest.

Here, we investigate whether neural networks can be effectively trained under this challenging setting using only weak supervision in the form of a small set of seed words, i.e., descriptive keywords for each aspect. For example, words like "price," "expensive," "cheap," and "money" are represen-

tative of the "Price" aspect. While a traditional aspect label is only associated with a single review, a small number of seed words can implicitly provide (noisy) aspect supervision for many reviews.

Training neural networks using seed words only is a challenging task. Indeed, we show that current weakly supervised networks fail to leverage the predictive power of the seed words. To address the shortcomings of previous approaches, we propose a more effective approach to "distill" the seed words in the neural network parameters. First, we present necessary background for our work.

## 2    BACKGROUND: NEURAL NETWORKS FOR ASPECT EXTRACTION

Consider a segment $s = (x_1, x_2, \ldots, x_N)$ composed of $N$ words. Our goal is to classify $s$ to $K$ aspects of interest $\{\alpha_1, \ldots, \alpha_K\}$, including the "General" aspect $\alpha_{\text{GEN}}$. In particular, we focus on learning a fixed-size vector representation $h = \text{EMB}(s) \in \mathbb{R}^l$ and using $h$ to predict a probability distribution $p = \langle p^1, \ldots, p^K \rangle$ over the $K$ aspect classes of interest: $p = \text{CLF}(h)$.

The state-of-the-art approaches for segment embedding use word embeddings: each word $x_j$ of $s$ indexes a row of a word embedding matrix $W_b \in \mathbb{R}^{V \times d}$ to get a vector representation $w_{x_j} \in \mathbb{R}^d$, where $V$ is the size of a predefined vocabulary and $d$ is the dimensionality of the word embeddings. The set of word embeddings $\{w_{x_1}, ..., w_{x_N}\}$ is then transformed to a vector $h$ using a vector composition function such as the unweighted/weighted average of word embeddings (Wieting et al., 2015; Arora et al., 2017), Recurrent Neural Networks (RNNs) (Wieting & Gimpel, 2017; Yang et al., 2016), and Convolutional Neural Networks (CNNs) (Kim, 2014; Hu et al., 2014). During classification (CLF), $h$ is fed to a neural network followed by the softmax function to get $p^1, \ldots, p^K$.

*Supervised* approaches use ground-truth aspect labels at the segment level to jointly learn the EMB and CLF function parameters. However, aspect labels are not available in our case. *Unsupervised* neural topic models avoid the requirement of aspect labels via autoencoding (Iyyer et al., 2016; He et al., 2017). In their Aspect Based Autoencoder (ABAE), He et al. (2017) reconstruct an embedding $h'$ for $s$ as a convex combination of $K$ aspect embeddings: $h' = \sum_{k=1}^{K} p^k A_k$, where $A_k \in \mathbb{R}^d$ is the $k$-th row of the aspect embedding matrix $A \in \mathbb{R}^{K \times d}$. The aspect embeddings $A$ (as well as the EMB and CLF function parameters) are learned by minimizing the segment reconstruction error.[1]

Unfortunately, unsupervised approaches like ABAE do not utilize information about the $K$ aspects of interest and thus the probabilities $p^1, \ldots, p^K$ cannot be used directly[2] for our downstream application. To address this issue, Angelidis & Lapata (2018) proposed a *weakly supervised* extension of ABAE. Their model, named Multi-seed Aspect Extractor, or MATE, learns more informative aspect representations by also considering a distinct set of seed words $G_k = \{g_{k1}, \ldots, g_{kL}\}$ for each aspect. In particular, MATE initializes the $k$-th row of the aspect embedding matrix $A$ to the weighted[3] average of the corresponding seed word embeddings: $A_k = \frac{1}{L} \sum_{l=1}^{L} z_{kl} w_{g_{kl}}$. As initializing the aspect embeddings to particular values does not guarantee that the aspect embeddings after training will still correspond to the aspects of interest, Angelidis & Lapata (2018) fix (but do not fine tune) the aspect embeddings $A$ and the word embeddings $W_b$ throughout training. However, as we will show next, MATE fails to effectively leverage the predictive power of seed words.

## 3    A DISTILLATION APPROACH FOR ASPECT EXTRACTION

In this work, we propose a weakly supervised approach for segment-level aspect extraction that leverages seed words as a stronger signal for supervision than MATE. Indeed, in contrast to Angelidis & Lapata (2018), who use average seed word vectors only for initialization, we use the individual seed words for supervision throughout the whole training process. Our approach adopts the paradigm of knowledge distillation (Hinton et al., 2015), according to which a simpler network

---

[1]Previous work models the reconstruction error using a contrastive max-margin objective (Weston et al., 2011; Pennington et al., 2014) as a function of $h$, $h'$, and the representations of randomly sampled segments.

[2]An alternative is to learn $K' > K$ topics and do a $K'$-to-$K$ mapping as a post-hoc step. However, this mapping requires either aspect labels or substantial human effort for associating topics with aspects.

[3]The weights $z_{kl}$ are estimated a priori using a variant of the clarity scoring function (Cronen-Townsend et al., 2002). This function requires aspect-annotated reviews, so Angelidis & Lapata (2018) use the validation set to estimate the seed weights.

(student) is trained to imitate the predictions of a complex network (teacher). After training, the parameters of the teacher will have been "distilled" to the parameters of the student and hopefully the student will perform comparably to the teacher for the task at hand. Our motivation is different: we can easily represent domain knowledge in simple and interpretable models but not in more complex neural networks. Thus, in our work, the teacher is a simple bag-of-words classifier that encodes the seed words, while the student is a more complex neural network, which is trained to distill the domain knowledge encoded by the teacher, as we describe below.

**Teacher: A bag-of-words classifier using seed words.** In Hinton et al. (2015), the teacher is trained on a labeled dataset. Here, we do not have training labels but rather seed words $G$ that are predictive of the $K$ aspects. Incorporating $G$ into (generalized) linear bag-of-words classifiers is straightforward: here, we initialize the weight matrix $W \in \mathbb{R}^{V \times K}$ and bias vector $b \in \mathbb{R}^K$ of a logistic regression classifier using the seed words:

$$W^{jk} = \begin{cases} \mathbb{1}\{j \in G_k\} & \text{if } \alpha_k \neq \alpha_{\text{GEN}} \\ -\mathbb{1}\{j \in (G_1 \cup \ldots \cup G_K)\} & \text{if } \alpha_k = \alpha_{\text{GEN}} \end{cases}, \qquad b^k = \begin{cases} 0 & \text{if } \alpha_k \neq \alpha_{\text{GEN}} \\ 1 & \text{if } \alpha_k = \alpha_{\text{GEN}} \end{cases} \qquad (1)$$

Under this weight configuration we consider seed words in an intuitive way: if at least one seed word appears in a segment, then the teacher assigns a higher score to the corresponding aspect than to $\alpha_{\text{GEN}}$, otherwise $\alpha_{\text{GEN}}$ gets the highest score among all aspects.[4] However, assigning hard binary weights to the teacher leads to ignoring the non-seed words, i.e., if a segment does not contain any seed words belonging to $G_k$, then the probability of the $k$-th aspect is zero. Of course, non-seed words can also be predictive for an aspect (especially given that we only consider a small, incomplete set of seed words). Next, we describe the architecture of the student network and how it can be trained to also consider non-seed words for aspect extraction.

**Student: An embedding-based neural network.** The student network is an embedding-based neural network: a segment is first embedded ($h = \text{EMB}(s)$) and then classified to the $K$ aspects ($p = \text{CLF}(h)$). For the EMB function we experiment with two choices: the unweighted average of word2vec embeddings and contextualized embeddings using BERT (Devlin et al., 2018).[5] For the CLF function we use the softmax classifier: $p_S = \text{softmax}(W_s h + b_s)$, where $W_s \in \mathbb{R}^{d \times K}$ and $b_s \in \mathbb{R}^k$ are the softmax classifier's weight and bias parameters, respectively. We train the student network to imitate the teacher's predictions by minimizing the cross entropy between the student's and the teacher's predictions.

Even if the teacher's predictions is the only supervision signal, the student can learn to generalize by associating aspects with non-seed words in addition to seed words. To encourage this behavior we use L2 regularization on the student's weights, and dropout on the word embeddings. To better understand the extent to which non-seed words can predict the aspects of interest, we experiment with completely dropping the seed words from the student's input. In particular, while the teacher receives the original segment during training, the student receives an edited version of the segment, where seed words belonging in $G$ have been replaced by an "UNK" id (like out-of-vocabulary words) and thus do not provide useful information for aspect classification. To imitate the predictions of the teacher, the student has to associate aspects with non-seed words during training.

## 4 EXPERIMENTS: ASPECT EXTRACTION IN PRODUCT REVIEWS

For training and evaluation, we use the OPOSUM dataset (Angelidis & Lapata, 2018), a subset of the Amazon Product Dataset (McAuley et al., 2015). OPOSUM contains Amazon reviews from six domains: Laptop Bags, Keyboards, Boots, Bluetooth Headsets, Televisions, and Vacuums. Aspect labels (for 9 aspects) are available at the segment-level[6] but only for the validation and test sets. For dataset details, see Angelidis & Lapata (2018). For a fair comparison, we use exactly the same 30 seed words (per aspect and domain) used in MATE.

In our experiments, we use exactly the same pre-processing (tokenization, stemming, and embedding) procedure as in Angelidis & Lapata (2018). For each domain, we train our model on the

---

[4]The symmetric case where all aspects classes are treated equally performs poorly (see Section 4).

[5]Here, we use the pre-trained BERT base uncased model (source: https://github.com/huggingface/pytorch-pretrained-BERT) and use average pooling to get segment embeddings.

[6]Segments in OPOSUM are defined as elementary discourse units (EDUs).

|  | L. Bags | Keyb/s | Boots | B/T H/S | TVs | Vac/s | AVG |
|---|---|---|---|---|---|---|---|
| ABAE | 38.1 | 38.6 | 35.2 | 37.6 | 39.5 | 38.1 | 37.9 |
| MATE-unweighted | 41.6 | 41.3 | 41.2 | 48.5 | 45.7 | 40.6 | 43.2 |
| MATE-weighted | 46.2 | 43.5 | 45.6 | 52.2 | 48.8 | 42.3 | 46.4 |
| MATE-weighted-MT | 48.6 | 45.3 | 46.4 | 54.5 | 51.8 | 47.7 | 49.1 |
| Teacher-BOW | 55.1 | 52.0 | 44.5 | 50.1 | 56.8 | 54.5 | 52.2 |
| Student-BOW | 57.3 | 56.2 | 48.8 | 59.8 | 59.6 | 55.8 | 56.3 |
| Student-W2V | 59.3 | 57.0 | 48.3 | **66.8** | **64.0** | 57.0 | 58.7 |
| Student-W2V-DSW | 51.3 | 57.2 | 46.6 | 63.0 | 62.1 | 57.1 | 56.2 |
| Student-BERT | **61.4** | **57.5** | **52.0** | 66.5 | 63.0 | **60.4** | **60.2** |

Table 1: Micro-averaged F1 reported for 9-class aspect extraction in Amazon product reviews.

training set without using any aspect labels, and only use the seed words $G$ via the teacher. We follow the same evaluation procedure as in Angelidis & Lapata (2018): we tune the hyperparameters on the validation set and report the micro-averaged F1 (9-class classification) in the test set averaged over 5 runs.

We compare the following models and baselines:

- **ABAE:** The unsupervised autoencoder of He et al. (2017), where the learned topics were mapped to the 9 aspects as a post-hoc step.
- **MATE-*:** The weakly supervised autoencoder of Angelidis & Lapata (2018). Various configurations include the initialization of the aspect matrix $A$ using the unweighted/weighted average of word embeddings and an extra multi-task training objective (MT).
- **Teacher-BOW:** A bag-of-words classifier with the weight configuration of Equation 1.
- **Student-*:** The student trained to imitate Teacher-BOW. Our experiments include bag-of-words (BOW) classifiers, the (unweighted) average of word2vec embeddings (W2V), and pre-trained BERT embeddings.

Table 1 reports the evaluation results for aspect extraction. The rightmost column reports the average performance across the 6 domains. MATE-* models outperform ABAE: using the seed words as a weak source of supervision leads to more accurate aspect predictions. Teacher-BOW uses the seed words but performs poorly. On the other hand, Teacher-BOW outperforms the MATE-* models: Teacher-BOW leverages the seed words effectively and in an intuitive way according to our knowledge about aspect extraction.

Student-BOW outperforms Teacher-BOW: the two models share the same architecture but regularizing the student's weights allows for non-seed words to also be considered for aspect extraction. The benefits of our distillation approach are highlighted using embedding-based networks. Student-W2V outperforms Teacher-BOW and Student-BOW, showing that obtaining segment representations as the average of word embeddings is more effective than bag-of-words representations for this task. Even when the seed words are not shown to the student during training (Student-W2V-DSW), it effectively learns to use non-seed words to determine the aspect, thus leading to more accurate predictions compared to the teacher.

Student-W2V outperforms the previously best performing model (i.e., MATE-weighted-MT) by 17.5%: although both models use exactly the same seed words, our distillation approach leverages the seed words more effectively for supervision than just for initialization. To demonstrate the simplicity and effectiveness of our approach, we do not use weights for the seed words nor a multitask objective for Student-W2V. Therefore, a fair comparison considers MATE-unweighted, which is outperformed by Student-W2V by 34.4%. Considering more sophisticated methods for segment embedding are promising to yield further performance improvements: Student-BERT achieves the best performance over all models.

In the future we plan to experiment with better methods for handling noisy seed words, and interactive learning approaches for learning better seed words.

**Acknowledgements:** This material is based upon work supported by the National Science Foundation under Grant No. IIS-15-63785.

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
