# OpenReview forum: "Training Neural Networks for Aspect Extraction Using Descriptive Keywords Only"
_ICLR.cc/2019/Workshop/LLD — LLD 2019_

### Official Review · AnonReviewer2 · 2019-04-05
**Convincing results, although the modelling could be improved**

**Rating:** 4
**Confidence:** 2

**Review:**

This paper presents an approach to classifying online reviews to aspect classes, when no gold labels for aspect labels are available. Their weakly supervised approach involves using a small set of seed words which describe these aspects. Their approach involves first showing that current weakly supervised approaches with neural networks are not suitable for learning the task under the setting used in the paper. To help with this issue, they turn to a variant of the paradigm of knowledge distillation, with the difference that they use a 'teacher' based on a bag-of-words classifier with seed words, and a student which is an embedding-based neural network.

In terms of the methodology used, the motivation for the simplicity of the student is somewhat lacking. While it is, of course, legitimate to leave more complicated formulations to future work, I am not certain of what exactly the authors mean when they find that that representations from the unweighted average of word embeddings is effective. Is the idea here simply that they tried a simple approach, and found the results to be sufficiently good to verify their approach?
It would have been interesting to see some more approaches here, in order to see whether improving the representations could yield better results, or whether perhaps the representations of the student do not matter so much. A relatively fast and simple thing to do, would be to embed the segments using a pre-trained BERT model.

Nonetheless, the results seem convincing, as their relatively simple distillation approach yields results which seem substantially above previous work.


Minor comments:
* The footnotes at ends of sentences should be written after punctuation

---

### Official Review · AnonReviewer1 · 2019-04-06
**Interesting but not entirely convincing**

**Rating:** 2
**Confidence:** 2

**Review:**

This submission addresses aspect extraction from product reviews by training a neural network (NN) under weak supervision.
This paper seems to me to be an extension of a previous work by Angelidis and Lapata (2018), who use a predefined set of seed words in order to extract aspect of interest from product reviews.
However, where Angelidis and Lapata fix both the aspect embedding matrix and the word embeddings for training, the authors in this work adopt the knowledge distillation paradigm by combining a bag-of-word model (as a teacher, simple model in which they can encode the domain knowledge), and an embedding-based NN (as a student, more complex but which can generalize better).

The reported results sound convincing, but I would have liked having a bit more discussion with regard to the two variants (BOW and EMB) of the Student model, as well as why, in most cases, dropping the seed words (SWD) does not help and when it does, it does not seem to be significant.
The authors would have strengthened their work by proving the claim that dropping/replacing seed words with the 'UNK' token actually teaches the Student network to associate aspects with non-seed words during training. My assumption is that it could simply be that the model associates aspects with another set of seed words (seed words which are not among the 30 seed words per aspect and domain from MATE, but which occur regularly, along side the UNK token).

* Few comments about the paper *
- The paper is easy to read;
- EMB and CLF abbreviations, although simple, are referred before being defined;
- the multi-task learning training objective of the MATE-* models is not introduced (nor why did they authors not considered it);

---

### Decision · Program_Chairs · 2019-04-16
**Acceptance Decision**

Accept